# Osteoblast Biocompatibility and Antibacterial Effects Using 2-Methacryloyloxyethyl Phosphocholine-Grafted Stainless-Steel Composite for Implant Applications

**DOI:** 10.3390/nano9070939

**Published:** 2019-06-28

**Authors:** Dave W. Chen, Hsin-Hsin Yu, Li-Jyuan Luo, Selvaraj Rajesh Kumar, Chien-Hao Chen, Tung-Yi Lin, Jui-Yang Lai, Shingjiang Jessie Lue

**Affiliations:** 1Department of Orthopedic Surgery, Chang Gung Memorial Hospital, Keelung City 401, Taiwan; 2College of Medicine, Chang Gung University, Guishan District, Taoyuan City 333, Taiwan; 3Department of Chemical and Materials Engineering and Green Technology Research Center, Chang Gung University, Guishan District, Taoyuan City 333, Taiwan; 4Graduate institute of Biomedical Engineering, Chang Gung University, Guishan District, Taoyuan City 333, Taiwan; 5Department of Radiation Oncology, Chang Gung Memorial Hospital, Guishan District, Taoyuan City 333, Taiwan; 6Department of Safety, Health and Environment Engineering, Ming Chi University of Technology, Taishan District, New Taipei City 243, Taiwan; 7R&D Center for Membrane Technology, Chung Yuan Christian University, Chung Li District, Taoyuan City 320, Taiwan

**Keywords:** 2-methacryloyloxyethyl phosphocholine (MPC), antibacterial activity, osseointegration, photo-induced polymerization, orthopedic implants

## Abstract

Poor osteogenesis and bacterial infections lead to an implant failure, so the enhanced osteogenic and antimicrobial activity of the implantable device is of great importance in orthopedic applications. In this study, 2-methacryloyloxyethyl phosphocholine (MPC) was grafted onto 316L stainless steel (SS) using a facile photo-induced radical graft polymerization method via a benzophenone (BP) photo initiator. Atomic force microscopy (AFM) was employed to determine the nanoscale morphological changes on the surface. The grafted BP-MPC layer was estimated to be tens of nanometers thick. The SS-BP-MPC composite was more hydrophilic and smoother than the untreated and BP-treated SS samples. *Staphylococcus aureus* (*S. aureus*) bacteria binding onto the SS-BP-MPC composite film surface was significantly reduced compared with the pristine SS and SS-BP samples. Mouse pre-osteoblast (MC3T3-E1) cells showed good adhesion on the MPC-modified samples and better proliferation and metabolic activity (73% higher) than the pristine SS sample. Biological studies revealed that grafting MPC onto the SS substrate enhanced the antibacterial efficiency and also retained osteoblast biocompatibility. This proposed procedure is promising for use with other implant materials.

## 1. Introduction

Microbial infection in implantable biomaterials due to biofilm formation is a major human infectious disease problem due to device-associated contamination. The most common bacterial pathogens associated with implantable infections are *Staphylococcus aureus* (*S. aureus*), which are found on wounds [1,2]. To inhibit microbial adherence and subsequent biofilm formation on implantable medical device surfaces is an effective strategy for preventing infections. Surface science contributes a critical role in the development of biomaterials because most of the biological reactions occur at their surfaces [3,4,5]. The optimal antibacterial implantable surface films required three important functions such as prevention of initial bacterial adhesion, killing all bacteria that succeed in overcoming this anti-adhesion barrier, and completely detached dead bacteria [6]. 

Among various implantable biomaterials, stainless steel (SS) has been broadly employed owing to its tremendous corrosion resistance, mechanical strength, and cost-effectiveness for various applications such as prostheses, orthopedic implants, and cardiovascular valves/stents [7]. The cell compatibility of the SS implants has been verified by clinical trials [8]. Nevertheless, the usage of pristine SS without surface modification is not suitable for implantable devices due to its unfavorable bactericidal response from proliferation, adhering, and forming a biofilm resistance [3]. For example, Fujii et al. observed that the pristine SS surface could have more bacterial adherence as well as biofilm formation, even though the addition of antibiotics in a pristine SS surface did not efficiently decrease the adhered bacteria number [9]. Therefore, active or passive SS substrate chemical modification is an efficient approach for improving surface properties because of the ability to control the interface and chemical flexibility [7]. Strategies such as plasma treatment [10], free radical copolymerization [9], photo-induced radical graft polymerization [11], thiol-ene click reaction [12], ion implantation [13], electro-grafting [14], etc., have been developed to coat/graft a suitable polymer or antibody derivative onto the SS surface. Recently Pang et al. demonstrated a SS substrate grafted with ionic liquid had improved antibacterial effects against gram-negative bacteria due to the dense passive layer and lipophilicity of imidazolium cation [12]. Xu et al. found that mussel-inspired terpolymers- and copolymers-coated SS surfaces efficiently reduced *Pseudomonas* sp. bacteria and microalgae adhesion compared with the pristine SS surface due to the minimum free energy at the polymer–water interface, high mobility, and hydrophilicity [15]. The prevention of both effective bacterial adhesion and deactivating bacteria growth upon contact with surface modified SS surfaces are of great interest and challenge in practical bio-implantation.

The synthetic bio-mimic 2-methacryloyloxyethyl phosphocholine (MPC) containing phospholipid polar side chain and the methacrylate group has the ability to repel protein adsorption and denaturation, prevent bacterial adherence, and exhibit excellent biocompatibility [16,17]. The MPC polymer was first established by the K. Ishihara group that applied for clinical usage of implantable artificial hips and artificial hearts [18,19,20]. MPC polymers have an admirable capability to inhibit cell adhesion, even once they originated a connection with entire blood cells or plasma in the absence of anticoagulants [21]. These characteristics are due to the electrically-neutral nature and hydrophilicity of the MPC, as well as the phosphorylcholine capability to induce a bulk-like structure with surrounding water molecules [19], which can incorporate 22 water molecules per MPC repeat unit [22,23]. Grafting MPC onto various substrates has been clinically employed with soft contact lenses, artificial lungs, artificial blood vessels and organs, intravascular guide wires, intravascular stents, and oxygenators under the approval of the Food and Drug Administration of the United States [24,25]. Kyomoto et al. thoroughly investigated MPC or poly(MPC) grafting onto various polymeric substrates (such as poly (ether ether ketone), cross-linked polyethylene (CLPE), vitamin E blended CLPE) to improve the unique properties of anti-protein adsorption, high lubricity, and low friction, good biocompatibility, cell membrane-like surface, and antibacterial adhesion effects for various orthopedic implantable devices [11,26,27]. Moreover, surface modified MPC polymers are known to exhibit noble in-vivo anti-biofouling effects with good blood and tissue compatibility [28,29,30]. The mouth-rinsing with MPC polymer clinical trial inhibited the growth of oral bacteria (*Streptococcus mutans*), confirming the prevention of dental plaque-related sicknesses [9]. The “superhydrophilicity” property of MPC-polymer-coated surfaces plays an important role in preventing microbial adhesion to oral epithelial cells and hydroxyapatite with suppressed biofilm formations [31]. Similarly, Zhang et al. demonstrated that a MPC-grafted dental resin composite could significantly reduce protein adsorption, bacterial attachment, and biofilm formation [32]. Most MPC-based literature extensively studied the prevention of biofilm formation and antibacterial effects, whereas fewer reports were available for cell compatibility and affinity to normal cells for implantable devices. 

Stainless steel (SS) is widely used for various biomedical implantable devices owing to its malleability and higher resistance to fatigue and corrosion [7]. The amphiphilic and surface roughness was a significant factor in the development of biocompatible materials for targeted implantable materials. In this work we report on biocompatible MPC polymer grafted onto a 316L SS substrate using the photo-induced radical graft polymerization method via benzophenone (BP), and investigate both antibacterial and cell survival effects. The physicochemical, contact angle, and surface properties of pristine SS, SS-BP, and SS-BP-MPC composites were investigated in detail. To investigate the favorable biomedical implantation, the in-vitro osteoblast biocompatibility, cell attachments, and antibacterial adhesion effects of composite films were studied and compared with the pristine SS sample. Our specific goal is to manipulate the SS composite surface properties for antibacterial adhesion effect but also to improve cell compatibility and survival for normal cells for promising implantable biomedical applications. 

## 2. Materials and Methods 

### 2.1. Materials 

AISI type 316L stainless steel (SS) with a size of 10 × 10 mm and thickness of 1 mm was purchased from Qiheng Stainless Steel Co., Ltd., Foshan, China. 2-Methacryloyloxyethyl phosphocholine (MPC, 99.9%), benzophenone (BP), sulfuric acid (H_2_SO_4_), hydrogen peroxide (H_2_O_2_), osmium tetroxide (OsO_4_), tween 80 (99%), paraformaldehyde (PFA), ethanol (99.9%), acetone (99%), lysogeny broth (LB), dimethylthiazol diphenyltetrazolium bromide (MTT) were purchased from Sigma-Aldrich, St. Louis, MO, USA. Dulbecco’s modified medium (DMEM) and fetal bovine serum (FBS) were obtained from Thermo Fisher Scientific Co., Ltd., Waltham, MA, USA. Crystal violet was purchased from Bioman Scientific Co., Ltd., Scottsdale, Arizona. Dimethyl hydrazine (DMSO) was purchased from Echo Chemical Co., Ltd., Toufen, Miaoli, Taiwan. Polystyrene tissue culture plates (TCPS) was purchased from Bio-fil, Santa Eulalia de Roncana, Barcelona.

### 2.2. Surface Polishing of 316L Stainless Steel (SS)

The 316L SS substrate was initially treated with tween 80 solution and washed with DI water. The SS substrate was then immersed into a mixed acidic solution of H_2_O_2_ and H_2_SO_4_ (1:3 ratio) and kept at ambient temperature for 1 h to polish the SS surface. The SS was then ultrasonically shaken with deionized (DI) water, ethanol, and acetone for 15 min to further remove the metal impurities or organic residue from the SS surface. The cleaned SS sample was purged with nitrogen (N_2_) gas and dried under vacuum oven at room temperature for 1 h.

### 2.3. Benzophenone Coating onto the SS Surface

Initially, 10 mg·mL^−1^ benzophenone (BP) was dissolved in the required amount of acetone. The dried SS substrate was then immersed in the solution for 1 min under a dark atmosphere, rinsed with DI water, and dried at room temperature [11] to obtain the BP coating onto the SS substrate. This sample was denoted as SS-BP. 

### 2.4. 2-Methacryloyloxyethyl Phosphocholine (MPC) Grafting onto SS-BP Surface

The MPC-grafted SS-BP substrate was prepared using the photo-induced radical graft polymerization method reported by Kyomoto et al. with some modification [17,26]. Briefly, 0.5 mol/L of MPC was dissolved in ethanol on a magnetic stirrer at room temperature. The SS-BP substrate was then immersed in an aqueous MPC solution. Following this, the sample was transferred into glass tubing and sealed tightly. This was placed in a water bath at 60 °C under a dark atmosphere. Afterwards, the sample was photo-irradiated using 365 nm ultraviolet (UV) light intensity of 5.0 mW·cm^−2^ for 90 min. After the polymerization, the substrate was removed from the reaction solution and glass tubing, and rinsed with DI water and ethanol to remove non-grafted polymer and unreacted monomer. Finally, the sample was primarily dried in N_2_ gas and then vacuum-dried for 1 h at room temperature to obtain the SS-BP-MPC composite.

### 2.5. Characterization

The surface microstructure of the pristine SS substrate, SS-BP, and SS-BP-MPC composite films was analyzed using a field emission scanning electron microscope (FESEM, SU8220, Hitachi, Tokyo, Japan) with an accelerating voltage of 5 to 15 kV. The surface topography of the pristine SS and its composite films were analyzed using atomic force microscope (AFM, Bruker, Billerica, MA, USA) via the tapping mode. The measurements were taken under dry conditions using the excitation frequency range in kHz, and the scan rate of 0.5 Hz with a tip velocity of µm/s. The chemical structure and functional groups of the pristine SS and its composites were analyzed from Fourier transform infrared (FTIR) spectrophotometer (Horiba FT-730, Minami-ku, Kyoto, Japan) equipped with an attenuated total reflection (ATR) sample holder at a wavenumber range of 4000–800 cm^−1^. The chemical composition of pristine and MPC-grafted SS surfaces were studied using X-ray photoelectron spectroscopy (XPS, K-Alpha, Thermo Fisher Scientific Inc., Walthan, MA, USA). The water contact angle (DM300, Kyowa Interface Science Co., Ltd., Saitama, Japan) of all samples was investigated using a drop shaped analysis method on the specimen surface. Three replicate measurements were tested on each sample and the averaged values were considered as the contact angles.

### 2.6. Antibacterial Adhesion Analysis

*Staphylococcus aureus* (*S. aureus*, ATCC: 25923) was one of the etiologic agents and most common human pathogen in fatal systematic disorders [25]. To investigate the antibacterial adhesion effect on the surface of pristine SS, SS-BP, and SS-BP-MPC samples, 1 × 10^5^
*S. aureus* bacterial cells with lysogeny broth (LB) medium were inoculated onto various samples in 12-well plates and incubated for 12 h at 37 °C. In literature, the susceptibility of *S. aureus* in mid- and late-exponential phase (6 and 12 h) and the stationary phase (24 h) to the antimicrobial agents found that the bacterial cells are most sensitive to the test substances in the late-exponential phase of culture [33]. As actively growing bacterial cultures become inactive, the cells tend to be more resistant and difficult to respond to antimicrobials, which may be unsuitable for use in the evaluation of antibacterial effects. Therefore, the present work selected the 12 h experimental condition (rather than other time intervals) for the incubation with bacterial cultures to more accurately investigate the antibacterial effects. Afterward, the soaked samples were taken out and the excess surface liquid removed with audition paper. Each sample was then dyed with 500 μL of 1% crystal violet for 15 min at ambient temperature. The stained samples were taken out and rinsed with phosphate buffer solution (PBS) three times, to wash out the surface crystal violet, and observed by photographic images followed by addition of 200 μL DMSO. In the case of quantitative analysis, the sample was placed into a Multiskan Spectrum Microplate Spectrophotometer (ThermoLabsystems, Vantaa, Finland). The data were then observed with an optical density (OD) at a wavelength of 620 nm. The adhered bacterial cell results were reported as the average value of measurements from three replications (n = 3). The *S. aureus* bacteria in the LB medium without any samples served as a control group.

### 2.7. Osteoblast Biocompatibility and Cell Proliferation

To investigate the osteoblast biocompatibility, the mouse pre-osteoblasts (MC3T3-E1, ATCC, American type culture collection) with a density of 5 × 10^4^ cells/mL were seeded in a 12-well tissue culture plate. It was cultured in regular growth medium consisting of a mixture containing Dulbecco’s modified medium (DMEM) and 10% fetal bovine serum (FBS) in the 5% CO_2_ incubator at 37 °C for 24 h. Before use in cell culture experiments, the samples were sterilized in a 70% ethanol solution and were thoroughly rinsed in sterilized PBS. In addition, all of the samples (the pristine SS, SS-BP, and SS-BP-MPC) were immersed in 500 μL of DMEM and incubated at 37 °C, 5% CO_2_ for 24 h as a test medium. Each test medium was then replaced with the original cell cultures medium and maintained at 37 °C in the presence of 5% CO_2_ for 48 h. The cells in regular growth medium, without contacting material samples, served as control groups. After that, the cultures were incubated with fresh culture medium with 0.5 mg/mL of dimethylthiazol diphenyltetrazolium bromide (MTT) reagent for 4 h followed by adding DMSO, and the data was analyzed at 570 nm using the Multiskan Spectrum Microplate Spectrophotometer.

For the cell proliferation part, all of the samples were sterilized and washed before being used. The MC3T3-E1 cellular morphology on culture of various SS substrates was investigated using FESEM [34]. Briefly, the SS substrate sample, after attaching the cells for 1, 4, and 7 days, was removed from the medium. It was then washed three times with PBS buffer, and 1 mL of 4% PFA was added and fixed at 4 °C for 8 h. Afterward the PFA was removed and washed three times with PBS buffer solution and then soaked for 10 min at 4 °C each time. One mL of OsO_4_ was then added under the dark room and immersed for 1 h at 4 °C. After removing OsO_4_, it was washed three times with deionized water. The alcohol gradient was dehydrated and washed three times with 50%, 75%, 80%, 95%, and 100%, respectively, and again soaked for 10 min at 4 °C each time. The sample was dried using CO_2_ supercritical drying, and gold ions were sputtered onto the surface with an ion sputter. The cell-attached SS substrate surface morphology was tested using FESEM. MC3T3-E1 cells in the control group without sample were placed in glass slides for morphological analysis. To quantify the cell attachment and proliferation, MTT assay was also used to determine various time points (1, 4, and 7 days), as describe in Section 2.7 (osteoblast biocompatibility part).

### 2.8. Statistical Analysis

All corresponding data were expressed as the mean ± standard deviation of three replicate measurements (n = 3). Multiple comparison tests were performed using the Tukey procedure using an add-on (MegaStat software) to Microsoft Excel.

## 3. Results and Discussion

The current work describes that MPC was grafted onto the SS surface using the photo-induced radical graft polymerization process and its antibacterial adhesion effects and cell compatibility were studied (as shown in Figure 1). Initially, the pristine SS substrate was passivated with piranha solution to oxidize the organic residues. The substrate was then cleaned with ethanol, acetone, and DI water. Piranha and protic solvents may induce hydroxyl or unsaturated metal oxides [35]. The phenyl moieties from BP units could readily react with the hydroxyl groups via condensation mechanism [12]. When BP is exposed to photo-irradiation by a UV-light source, photo-induced scission or pinacolization reactions are induced. This results in the formation of ketyl radicals that perform as photo-initiators [17]. Hence, BP moieties have been extensively used as a photo-initiator to stimulate chemical conjugation [36]. The free radicals from UV-irradiation, a BP unit in SS undergo the surface-initiated graft polymerization reactions in the methacrylate-type monomer (e.g. MPC) aqueous solution. This facilitates the direct or self-initiated grafting of MPC onto the SS-BP surface in the presence of a BP photo-initiator, thereby resulting in the formation of SS-BP-MPC composite. Thereby, the benefits of photo-induced radical graft polymerization in MPC polymer produce a thin coating on the surface and had negligible effects on the bulk properties of the SS substrate. Retaining their SS substrate properties were also important in commercial clinical usage.

### 3.1. Surface Analysis

The surface microstructure of the SS, SS-BP, and SS-BP-MPC composites was analyzed using a field emission scanning electron microscope (FESEM). The pristine SS substrate had more cracks with a rough surface, as shown in Figure 2a. With the addition of BP or MPC on the SS substrate (Figure 2b,c), the film surface did not display significant appearance changes. It is speculated that the thickness (nanometer-scaled MPC layer on the SS surface) of the grafted layer was not sufficient to affect the surface. 

To further investigate the detailed surface topographic changes in the SS-BP and SS-BP-MPC composites compared with pristine SS, atomic force microscopy (AFM) was employed to determine nanoscale morphological changes. The topographic AFM image of the pristine SS substrate exhibited rough surfaces with the surface root-mean-square (RMS) roughness (Rq) of 42.8 nm, Figure 3a. Upon the BP coating on the SS substrate, the surface roughness was gradually reduced, and the Rq value was decreased to 38.6 nm (Figure 3b). In contrast, the MPC-grafted surface (SS-BP-MPC) showed a rather smooth and dense morphology (Figure 3c) with Rq values between 2.31 and 1.45 nm. Kyomoto et al. [11] showed that the grafting of PMPC onto a vitamin E-blended cross-linked polyethylene substrate could significantly reduce the Rq value from 5.29 to 0.71 nm, resulting in a smoother surface. Similarly, the present work confirms the smooth surface with reduced Rq values in the SS-BP-MPC composite due to its successful thin MPC layer grafting. Moreover, neither delamination nor cracking was detected on the SS-BP-MPC composite surface, which further confirms the successful MPC layer grafting. The FESEM images show a resolution of ~10 µm and the macrostructure does not seem to differ between samples. The AFM reveals the µm resolution and the surface morphology is smoother, especially on the MPC-modified sample. This result shows the submicron or nanomicron topological changes upon MPC grafting. 

### 3.2. Structural and Compositional Analysis

The functional group and chemical structure of the pristine SS, SS-BP, and SS-BP-MPC composites were studied using ATR-FTIR, as represented in Appendix A. The pristine SS surface had functional groups of O–H, C–O, and C–H stretching vibrations. Compared with the pristine SS sample, the SS-BP and SS-BP-MPC composite additionally had a broad peak at 3080 cm^−1^, attributed to the OH− group of hydroxyl molecules. No other significant characteristic peaks were observed after grafting BP and MPC onto the SS substrate. Based on the ATR-FTIR mode, the corresponding functional groups of MPC were not clearly visible due to thin layer of hydrophilic MPC grafting effects. Lin et al. reported that the sensitive MPC peaks were visible using X-ray photoelectron spectroscopy (XPS) analysis [36] but not with FTIR. Therefore, XPS was used to further investigate the in-depth grafting of MPC onto the SS substrate.

XPS was used to investigate the atomic composition of pristine SS and the thin coating sample SS-BP-MPC. The full scan spectra are shown in Figure 4a,b. The pristine SS elemental signals show carbon (C 1s), oxygen (O 1s), chromium (Cr 2p), iron (Fe 2p), and nickel (Ni 2p), respectively, as shown in Figure 4a. All the corresponding XPS peaks matched with the reported literature [10,15]. In the full scan, the carbon content in pristine SS was higher due to predictable adventitious hydrocarbon contaminants [3]. Upon grafting BP-MPC onto the SS substrate, additional nitrogen (N 1s) and phosphorous (P 2p) elemental peaks were present on the SS-BP-MPC sample surface (Figure 4b). The metallic signals were almost invisible, indicating the grafted BP-MPC layer was able to block XPS beam penetration. Therefore, the BP-MPC layer was estimated to be tens of nanometers thick [16,37,38]. Moreover, the relative population of O 1s groups increased whereas the C 1s peak intensity decreased. This signals the surface modification has increased hydrophilicity in the SS-BP-MPC sample. All other corresponding SS-BP-MPC peaks were similar to the pristine SS sample and experienced negligible changes.

The high resolution spectra of N 2p and P 2p peaks obtained from the pristine SS and SS-BP-MPC surfaces are displayed in Figure 4c–f. The SS-BP-MPC composite presented a significant P 2p signal at the binding energy of 134 eV and an N 1s peak at 401 eV (Figure 4d,f), whereas the pristine SS sample showed no corresponding peaks (Figure 4c,e). These two peaks are recognized for protonated phosphate and ammonium functional groups in MPC units grafted onto the SS surface and matched the reported literature [39,40]. In the composite, the nitrogen and phosphorus elemental contents were increased to 3.67 and 4.61 wt%, respectively, which were undetected in the pristine SS sample. This confirmed that the three hydrophobic methyl groups, bonded to nitrogen in phosphorylcholine from MPC, are surface exposed and closely linked by electrostatic force [19]. Although, the ATR-FTIR spectrum did not show the characteristic peaks due to weak reflective signals from MPC on the SS surface, whereas the more sensitive XPS analysis revealed P 2p and N 1s peaks, confirming successful MPC grafting onto the SS substrate.

### 3.3. Water Contact Angle Measurements

The water contact angle analysis was employed to investigate the interface wetting behavior of the pristine SS substrate, SS-BP, and SS-BP-MPC composites. The water contact angle of the pristine SS surface was 80.67° ± 2.08° (Figure 5) demonstrating a hydrophobic surface. When the BP was treated onto the SS substrate, the contact angle decreased to 45.33° ± 2.52°. This BP addition greatly enhanced the surface hydrophilicity of the SS-BP film. This was due to the fact that the liquid droplet on the SS-BP surface film could have a balance between the attractive force of internal molecules (cohesive force) and liquid molecules (adhesive force) to the surface [36]. Moreover, the water contact angle was further reduced (35.67° ± 0.58°) by MPC grafting onto the SS-BP surface. Kyomoto et al. [11] reported that grafting PMPC onto HD-CLPE (VE) resulted in a reduced water contact angle to 26.0° from 90.8°. It is speculated that the ammonium and phosphate functional groups of MPC units significantly increased the hydrophilicity of the composite film. The zwitterionic MPC groups not only interact with water molecules via hydrogen bonding but also induced hydration layer formation by electrostatic binding energy [16]. Therefore, the hydrophilicity of the SS-BP-MPC composite was greatly improved over other samples. 

However, our MPC-modified sample was not as hydrophilic as reported in the literature (<26° in [26,41,42]). We suspect that the sample preparation and drying condition made the MPC chains collapse, resulting in a rather smooth surface (as shown inserted schematic diagram in Figure 5). At the same time, part of the hydrophobic segment (shown in orange color of Figure 5) was laid on the SS surface and was exposed to air. This configuration maintains the MPC’s property of amphiphilic nature. Therefore, the water contact angle was at a moderate 36°.

### 3.4. Antibacterial Test

Crystal violet was used as a stain for *S. aureus* to investigate the qualitative analysis of the antibacterial effects. Figure 6a shows that the pristine SS was entirely dispensed with purple stains on the surface, indicating that SS did not have the ability to resist bacterial attachment. When BP was treated onto the SS surface (Figure 6b), the distribution of purple stains slightly decreased and formed aggregations on the surface. This indicates that a change in the surface energy of the BP film may resist the bacterial interaction. Further grafting of MPC onto the SS-BP surface resulted in remarkably reduced bacterial attachments, as shown in less purple staining in Figure 7c. Owing to MPC’s high affinity for water molecules, a strong hydration layer could be formed [43]. Such hydration layer could strongly reduce the surface roughness [11] and co-efficient of friction [26]. Thereby, the MPC layer improves anti-adhesive force between the bacteria and MPC surface, thereby resisting biofilm formation [44]. The chemical composition, surface topography, and surface energy of the SS-BP-MPC film imply a crucial role of antibacterial ability and biofilm resistance. 

The quantitative results from the SS, SS-BP, and SS-BP-MPC samples were analyzed by suspension absorption values incubation with *S. aureus* bacteria and compared with the control group, as shown in Figure 7. The surface residual bacteria amount on the pristine SS and SS-BP films were about 87% and 84%, respectively, compared with the control sample. Thereby, the antibacterial adhesion effect of SS and SS-BP films were only 13% and 16%, respectively, which indicated low ability to resist bacterial attachment. Interestingly, the surface residual bacteria amount on the SS-BP-MPC film was reduced to 66%, and the antibacterial adhesion effect could increase to 34%. It was confirmed that MPC has the ability to inhibit the adhesion of *S. aureus* bacteria on the surface with high antibacterial effect. The amount of bacterial adhesion on the SS-BP-MPC composite was significantly reduced by 2.5 fold compared with pristine SS due to lower surface roughness and greater hydrophilicity. Kyomoto et al. [11] demonstrated that the MPC grafted substrate has good anti-adhesion of *S. aureus* bacteria compared with the untreated sample due to the smooth surface and hydrophilicity of MPC units. The MPC’s zwitterionic nature forms a hydration layer on the surface via hydrogen bonding and electrostatic interaction [45], which may contribute to the strong repulsion to *S. aureus* bacteria. 

### 3.5. Osteoblast Biocompatibility

To assess whether the surface coating is suitable for use as biomaterials, an indirect method was used to assess cell biocompatibility. The sample was soaked in MC3T3-E1 cells for 24 h and the cell population metabolic activity was assessed using the MTT assay. The osteoblast biocompatibility of the pristine SS, SS-BP, and SS-BP-MPC were compared with the TCPS control group, as shown in Figure 8. The cell compatibility of the SS-BP-MPC composite was about 96% and the values were similar to the control group. This phenomenon might be due to the presence of surface-enriched biocompatible phosphorylcholine groups from the MPC moieties, which have favorable interaction with the cell surface. Conversely, the cell compatibility of the SS-BP film was reduced (84%) as compared with the TCPS control sample. This suggests that the presence of BP on the SS surface was less favorable for cell survival. The cell biocompatibility of all samples was higher than 80%, although the SS-BP group was lower than other groups, but still within acceptable limits for implantable devices [46].

### 3.6. In-Vitro Osteoblast Proliferation 

The in-vitro MC3T3-E1 cell proliferation was evaluated on the pristine SS, SS-BP, and SS-BP-MPC composite surfaces with different cultivation periods (one, four, and seven days). In the first day, the cell proliferation on the pristine SS surface was significantly smaller than those of the SS-BP and SS-BP-MPC composites (Figure 9). This was owing to greater hydrophobicity and rougher surface on the pristine SS substrate (as shown in AFM). For the SS-BP and SS-BP-MPC composites, the cell attachment levels were increased to 22% and 53%, respectively, compared with the pristine SS substrate. When incubated for four days with cell culture, the active cell proliferation onto SS-BP-MPC was significantly increased. The cell metabolic activity on the SS-BP-MPC was 73% higher than that on the pristine SS sample. Increasing cell proliferation could further confirm good stability and favorable interaction with biomolecules. After seven days of culture, we found that the OD value of MC3T3-E1 cells on SS-BP-MPC was lower than that with four days’ cell culture. Xu et al. [47] investigated long-term observation (four and eight days) of cell viability and its morphology using MPC composite-based hydrogels. Their results demonstrated that the cell activity at four days was higher than that at eight days. This was due to evaporation leading to water loss from the hydrogel, which adversely altered the cell microenvironment. The present work speculates that after four days the proliferation reaches a plateau and the cells then begin to differentiate into osteoblasts.

MC3T3-E1 cells were cultured onto various samples. The metabolic activities of the total cell population after four days were examined using FESEM analysis. Micrographs of the cell attachment on the pristine SS, SS-BP, and SS-BP-MPC composites are shown in Figure 10a–c. The MC3T3-E1 cell attachment ratio on the pristine SS was low as there were no functional groups on the SS surface. In the SS-BP and SS-BP-MPC cases, the good cell attachment onto the surface was clearly visible and the cell morphology was similar to the control (Figure 10d). The result indicates the BP-MPC layer had osteoblast biocompatibility and improved biological activity.

The hydrophilic MPC-based polymers are able to suppress non-specific binding proteins [48,49]. The resistance to HepG2 cell adhesion is also reported [50]. It is interesting to note, in this work, the cell MC3T3-E1 cell attachment and growth on BP- and BP-MPC-modified samples were improved as compared with SS substrate. This finding is not extraordinary, as Katayama et al. [51] also observed the same trend on the MPL copolymer-modified SS substrate, although they did not give sufficient explanation or proof. On further examination of our sample characteristics we found out the MPC-grafted sample had a moderate contact angle (~36° vs. <26° [17,19,41,42]). The rather flat surface (as shown in inserted Figure 5), indicating it was not a highly-grafted treatment [11,43,51]. The collapsed BP-MPC chains (as discussed in Section 3.3) had suitable configuration to expose both hydrophilic and hydrophobic domains (Figure 1a), which allows the hydrophilic part to prevent bacterial growth and the hydrophobic segment for cell interaction. Similarly, Seo et al. demonstrated that the heterogeneously-segregated hydrophobic domains could have noteworthy effects on cellular response [52]. Iwasaki et al. also demonstrated that the hydrophilic:hydrophobic length ratio had a dramatic effect on adherent L929 cells [35]. Our data indicates that antibacterial and cell adhesion properties may be achieved (as shown in Figure 1b) simultaneously if the BP-MPC layer is tuned to an amphiphilic structure.

## 4. Conclusions

A facile photo-induced radical graft polymerization method was employed in this study to prepare a SS-BP-MPC composite using benzophenone as a photo-initiator. The thin and smooth MPC layer on the SS substrate was clearly visible in AFM topography. This thin MPC layer on the composite decreased the surface roughness (Rq) from 42.8 to 1.45 nm compared with the pristine SS sample. The chemical structure and binding composition of the SS-BP-MPC composite were determined using XPS. The MPC is a high water-wettability compound, thereby rendering a hydrophilic surface onto the SS-BP-MPC composite (with water contact angle of 36°, lower than 80° of the pristine SS). The above results confirmed the successful grafting of MPC onto the SS surface. The SS-BP-MPC sample exhibited 34% antibacterial effect and good cell biocompatibility. After four days of incubation, the MC3T3-E1 cell activity on the SS-BP-MPC composite was 73% higher than on the pristine SS sample. Our findings confirm that MPC treatment is not only a favorable tool for the prevention of infectious diseases and microbial inhibition but also improves osteoblast cell attachment and proliferation. The MPC modification process could possibly be applied as an active surface design for commercial antibacterial adhesion and an alternative clarification for various implantable medical devices with cost-effective benefits.

## Figures and Tables

**Figure 1 nanomaterials-09-00939-f001:**
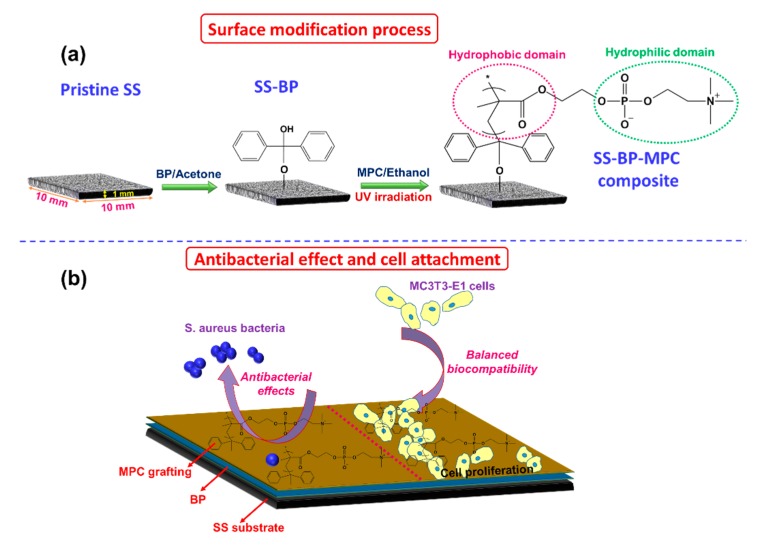
Schematic diagram of benzophenone and 2-methacryloyloxyethyl phosphocholine grafted stainless steel (SS-BP-MPC) composite for (**a**) surface modification process and (**b**) antibacterial effects and cell attachment effects.

**Figure 2 nanomaterials-09-00939-f002:**
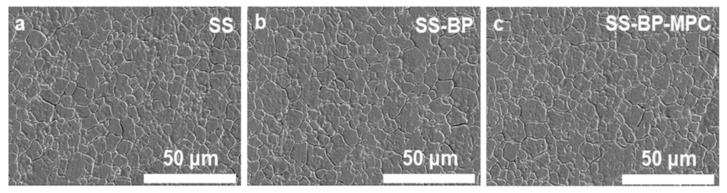
Field emission scanning electron microscopic (FESEM) images for (**a**) pristine SS substrate, (**b**) SS-BP, and (**c**) SS-BP-MPC composites.

**Figure 3 nanomaterials-09-00939-f003:**
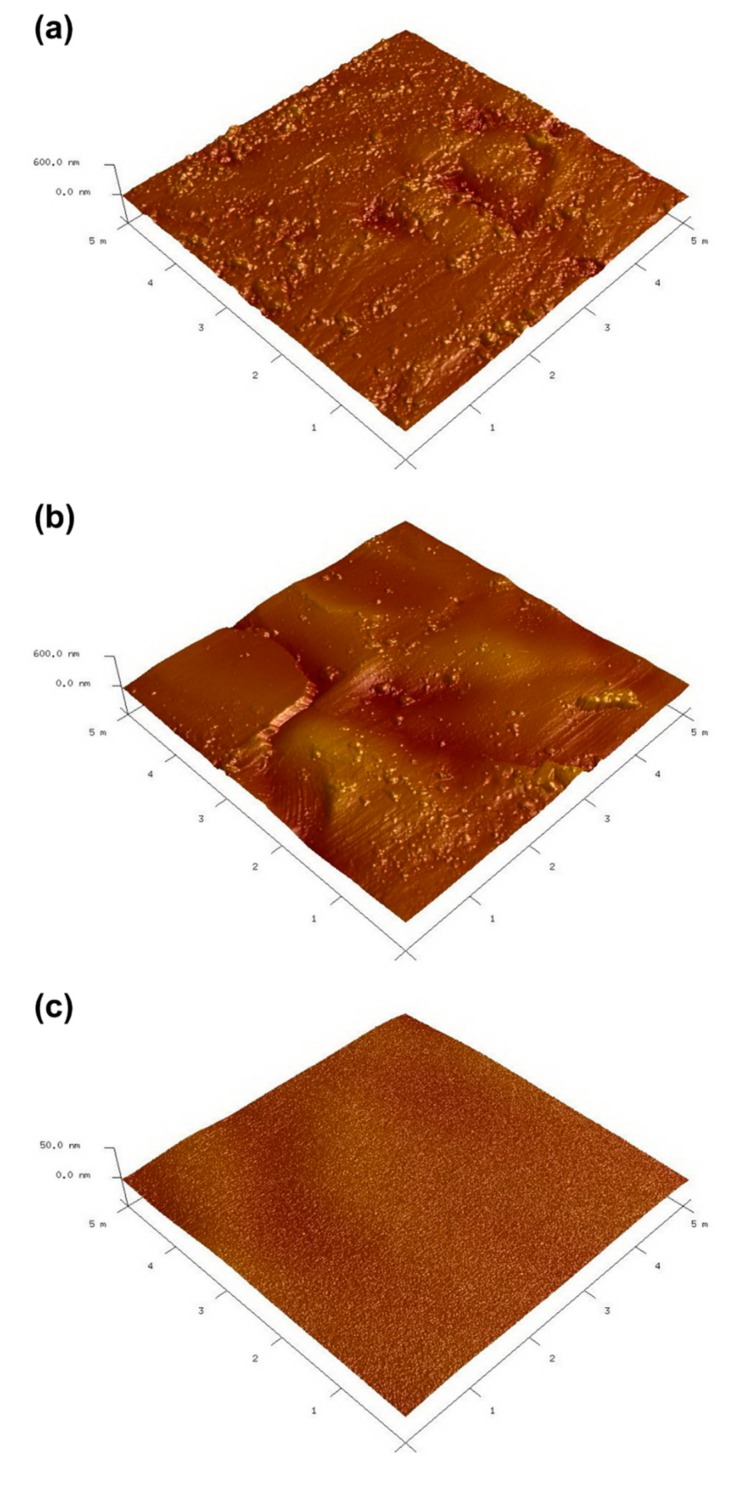
Surface topographic micrographs for (**a**) pristine SS substrate, (**b**) SS-BP, and (**c**) SS-BP-MPC composites.

**Figure 4 nanomaterials-09-00939-f004:**
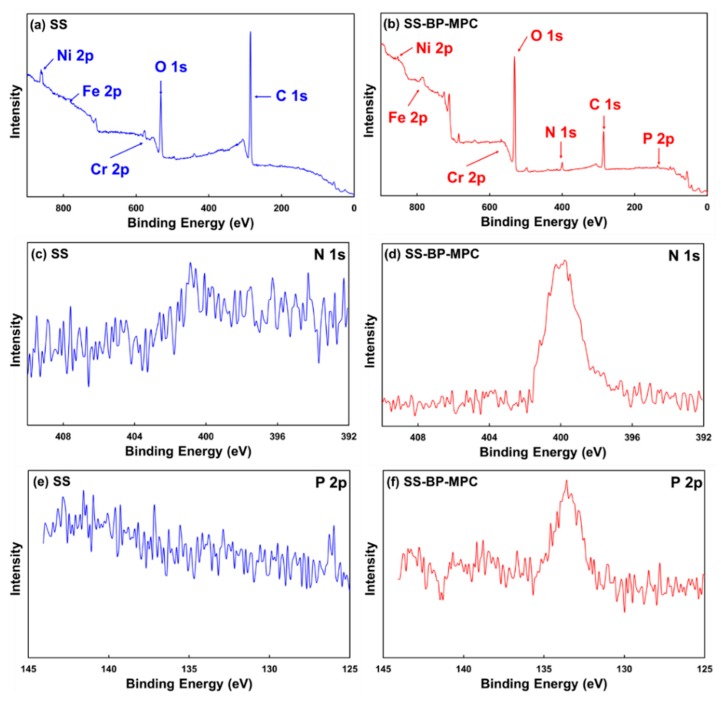
X-ray photoelectron spectroscopic (XPS) analysis for (**a**,**b**) full scan of pristine SS substrate and SS-BP-MPC composite, (**c**,**d**) N 1s spectra of pristine SS substrate and SS-BP-MPC composite, and (**e**,**f**) P 2p spectra of pristine SS substrate and SS-BP-MPC composite, respectively.

**Figure 5 nanomaterials-09-00939-f005:**
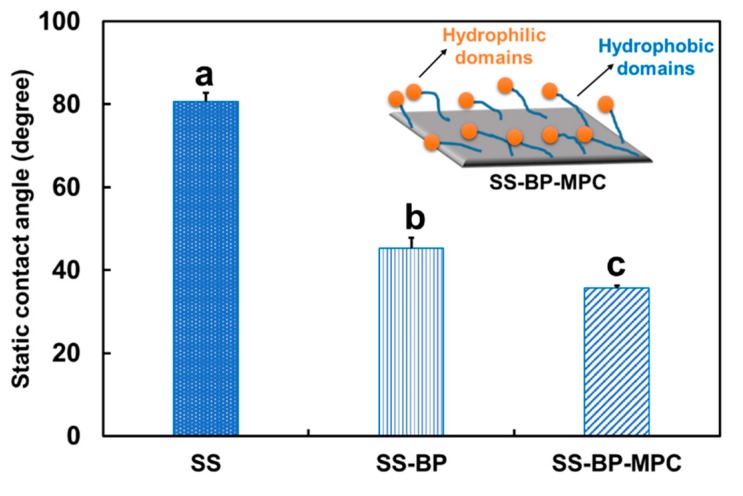
Static water contact angles for pristine SS, SS-BP, and SS-BP-MPC composites (n = 3). Data with different letters mean they are significantly different on multiple comparison at α = 0.05 using the Tukey test. The inserted diagram shows a schematic diagram of the collapsed MPC chain on the substrate surface.

**Figure 6 nanomaterials-09-00939-f006:**
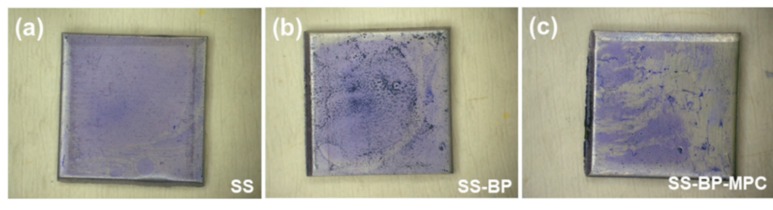
Photographs of (**a**) pristine SS, (**b**) SS-BP, and (**c**) SS-BP-MPC composites co-cultured with *S. aureus* bacteria for 12 h.

**Figure 7 nanomaterials-09-00939-f007:**
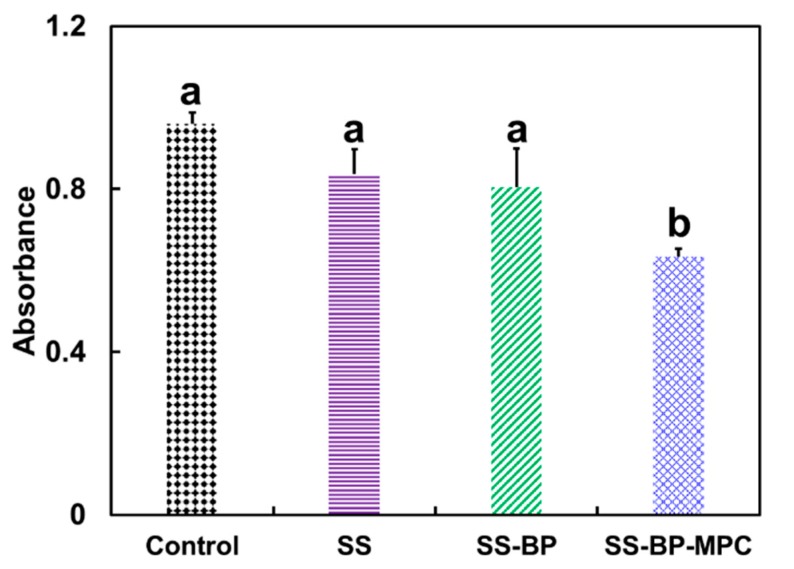
Quantitative analysis of bacterial adherence onto the surface of pristine SS, SS-BP, and SS-BP-MPC composites co-cultured with *S. aureus* for 12 h (n = 3). The *S. aureus* bacteria in the lysogeny broth (LB) medium without any samples served as a control group. Data with different letters mean they are significantly different on multiple comparison at α = 0.05 using the Tukey test.

**Figure 8 nanomaterials-09-00939-f008:**
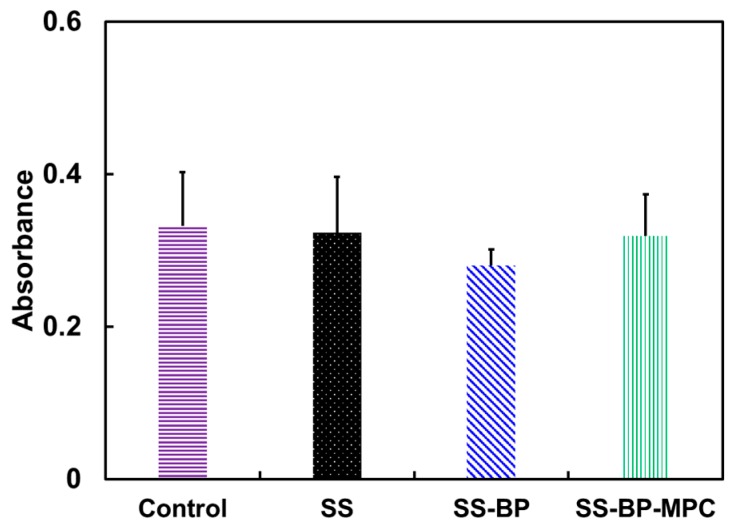
Cytotoxicity of the MC3T3-E1 cell line treated with control, pristine SS, SS-BP, and SS-BP-MPC composites using the dimethylthiazol diphenyltetrazolium bromide (MTT) assay (n = 3). The cells in regular growth medium, without contacting material samples, served as control groups. The statistical analysis indicates the means are not significantly different at α = 0.05.

**Figure 9 nanomaterials-09-00939-f009:**
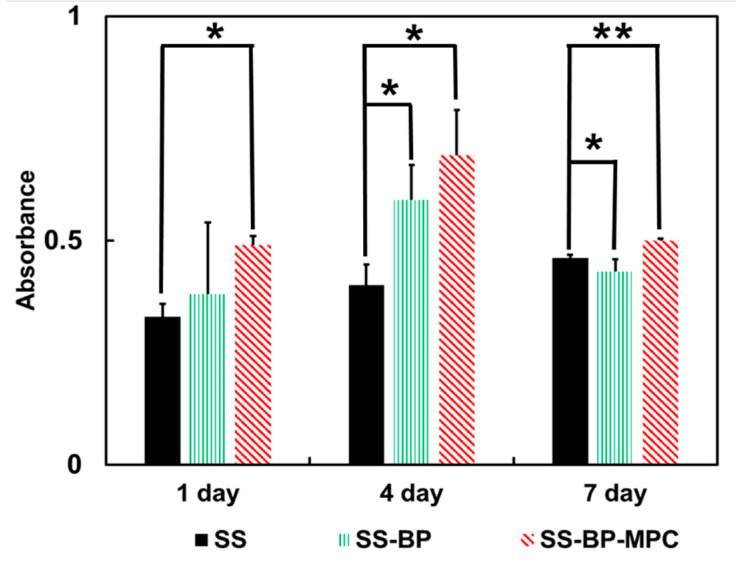
MC3T3-E1 cell growth on the surface of pristine SS, SS-BP, and SS-BP-MPC composites with various time intervals using the MTT assay (n = 3). Data with asterisks mean they are different from the SS sample at α = 0.05 within the same day group.

**Figure 10 nanomaterials-09-00939-f010:**
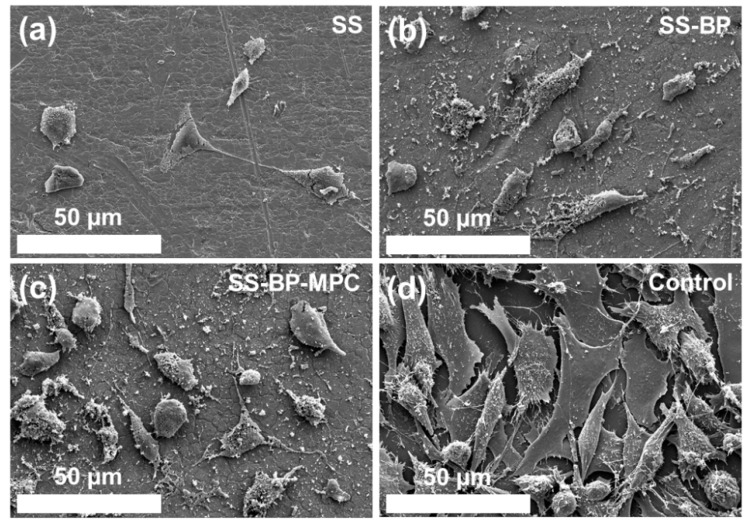
FESEM micrographs for MC3T3-E1 cells co-cultured on surfaces of (**a**) pristine SS, (**b**) SS-BP, and (**c**) SS-BP-MPC composites for four days and compared with (**d**) control group.

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
