# Peer review of "Osteoblast Biocompatibility and Antibacterial Effects Using 2-Methacryloyloxyethyl Phosphocholine-Grafted Stainless-Steel Composite for Implant Applications"

_nanomaterials, 2019, doi:10.3390/nano9070939_

Round 1
Reviewer 1 Report
MPC was originally developed by Prof.Ishihara. You should cite some review articles for MPC by Prof.Ishihara.
What is the size and shape of your SS samples?
Please add detailed conditions for characterization measurements, such as accelerating voltage for SEM, probes, resonance frequencies for AFM etc.
What is the level of significance in the statistical analysis and what is the software for statistics?
How does BP bind to SS surface?
You mentioned BP film was produced on SS surface. But BP is not polymer and I think BP film cannot be produced on SS surface.
What is the bond strength or durability of MPC layer?
The contact angle approximately 30-40 degrees is no super hydrophilic. I think your result does not accord with a result of reference No. 23.
You only tested S.aureus. You should check other bacteria.
Please put the title in Y-axis in Figure 5, 8 and 9.
Author Response
Reviewer 1
English language and style
( ) Extensive editing of English language and style required
( ) Moderate English changes required
( ) English language and style are fine/minor spell check required
(x) I don't feel qualified to judge about the English language and style
Comments and Suggestions for Authors
Q1) MPC was originally developed by Prof. Ishihara. You should cite some review articles for MPC by Prof. Ishihara.
Response: We agree with the Reviewer’s facts. We already cited many articles based on Prof. Ishihara et al. groups. We included more review articles and a book chapter (Ref. 18-20) in the 3rd paragraph of the introduction section, p. 2.
Q2) What is the size and shape of your SS samples?
Response: The SS sample size is 10 mm × 10 mm × 1 mm (L×b) with a thickness of 1 mm with a square shape. The details are included in section 2.1, p. 3.
Q3) Please add detailed conditions for characterization measurements, such as accelerating voltage for SEM, probes, resonance frequencies for AFM etc.
Response: The details are included in section 2.5, pp. 4-5.
Q4) What is the level of significance in the statistical analysis and what is the software for statistics?
Response: The level of statistical significance is α=0.05. We used MegaStat software (a MS Excel add on) to calculate the significant difference in multiple comparisons. The details are included in section 2.8, p. 6.
Q5) How does BP bind to SS surface?
Response: Initially, the pristine SS substrate was passivated with piranha solution to oxidize the organic residues. The substrate was then cleaned with ethanol, acetone and DI water. Piranha and protic solvents may induce hydroxyl or unsaturated metal oxides. [ACS Applied Materials and Interfaces 4 (2012) 3254-3260]. The phenyl moieties from BP units could readily react with the hydroxyl groups via condensation mechanism [Colloids and Surfaces B: Biointerfaces 126 (2015) 162.168]. The pristine SS surface had functional groups of O-H, C-O, C-H stretching vibrations, as shown in the FTIR spectra (supplementary information Fig. S1). The functional groups on the SS surface may possibly be responsible for the substrate-binding attribute. The details are included in the 1st paragraph of section 3 (p. 6) and section 3.2 (p. 9).
Q6) You mentioned BP film was produced on SS surface. But BP is not polymer and I think BP film cannot be produced on SS surface.
Response: Please see the response to question Q5.
Q7) What is the bond strength or durability of MPC layer?
Response: The MPC layer durability was not studied in the present work. We believe the MPC layer was stable in dry condition. More experimental analysis is needed to determine MPC layer stability under different environmental and/or biological conditions.
Q8) The contact angle approximately 30-40 degrees is no super hydrophilic. I think your result does not accord with a result of reference No. 23.
Response: Thank you for your valuable suggestion. The References and the sentence in the 2nd paragraph of section 3.4, p. 12 were revised.
Q9) You only tested S. aureus. You should check other bacteria.
Response: The authors deeply appreciate the Anonymous Reviewer’s constructive comments. However, every single experiment involving the use of biological hazards (can cause adverse health effects) should be carefully checked and examined by the ethics committee from the school’s office of environmental safety and hygiene. This administrative and academic process is usually time-consuming and it is difficult to quickly acquire committee approval during the revision of our work at this time. The Reviewer’s professional guidance is highly appreciated. The authors definitely will keep this issue in mind in the experimental design of future studies.
Q10) Please put the title in Y-axis in Figure 5, 8 and 9
Response: Y-axis title is revised in the Fig. 4, 7, 8 and 9.

Reviewer 2 Report
This manuscript describes surface modification of SS substrate by photoinduced graft polymerization of MPC. Surface characterization and biological examination were carried out. Many data were conjugated in the manuscript, however, some were not important to make discussion. Also, most important point was that the authors should explain the chemical reaction between BP layer and SS substrate. Therefore, major revisions should be needed before accepting the manuscript for publication.
1) In abstract, P1L29: the contents of nitrogen and phosphorus are not important. By the way, were the values in wt %?
2) In Page 3 L125: The surface of the SS substrate did not have any contamination? This description indicated that the no organic components on the surface. However, in Fig. 5, the bear SS substrate had clear signal attributed to carbon. Clear description will be needed.
3) When photoirradiation was carried out, BP layer may be activated and radicals were formed. However, no reaction might be occurred with clean SS substrate. What kind of binding reaction did occur? From viewpoint of this, Figure 1a could not be acceptable.
4) As Figure 2 and Figure 3 are not considerable. The reviewer recommended that these figures would be deleted. The thickness of the PMPC layer should be added instead of RMS value. Did authors consider that under dry condition, did the PMPC layer collapse or stand up on the surface?
5) Figure 4 is not clear indication. It should move to the supporting data.
6) As biological test, bacteria adhesion and cell adhesion were carried out. However, the effects of the PMPC layer on the substrate did not observed significantly. In generally, many reports have been indicated that the PMPC layer can prevent both bacteria adhesion and cell adhesion. Of course the reviewer understand the mode of binding reaction with substrate was different. However, if the surface was covered with the PMPC layer stably, the resistance performance for bacteria and cell adhesion should be the same with the published data. One possibility is unstable of the PMPC layer during culturing period. That is, the reviewer has already pointed out, the binding between BP layer and SS substrate was too weak to maintain the PMPC layer for long term. The authors must make sufficient explanation about this point.
Author Response
Reviewer 2
English language and style
( ) Extensive editing of English language and style required
(x) Moderate English changes required
( ) English language and style are fine/minor spell check required
( ) I don't feel qualified to judge about the English language and style
Response: The language is revised in the entire manuscript.
Comments and Suggestions for Authors
This manuscript describes surface modification of SS substrate by photoinduced graft polymerization of MPC. Surface characterization and biological examination were carried out. Many data were conjugated in the manuscript, however, some were not important to make discussion. Also, most important point was that the authors should explain the chemical reaction between BP layer and SS substrate. Therefore, major revisions should be needed before accepting the manuscript for publication.
Thank you very much for your positive response to our work. Based on your suggestion, we revised the manuscript. We hope you will support these changes.
Q1) In abstract, P1L29: the contents of nitrogen and phosphorus are not important. By the way, were the values in wt %?
Response: Thank you for your suggestion. The sentence is revised in the abstract, p.1. The wt% values of Nitrogen and Phosphorous is included in the 3rd paragraph of the section 3.2, p. 10.
Q2) In Page 3 L125: The surface of the SS substrate did not have any contamination? This description indicated that the no organic components on the surface. However, in Fig. 5, the bear SS substrate had clear signal attributed to carbon. Clear description will be needed.
Response: The as received 316L SS sample had more carbon contamination (>43 wt%). In the present work, the acid dipping was systematically applied to the pristine SS sample to reduce the carbon contamination to 18.95 wt% (based on XPS analysis). Similar carbon contamination level in SS substrate after acid treatment was also reported by Haidopoulos et al. [J Mater Sci: Mater Med. 17 (2006) 647–657] for further surface modification and usage of medical devices. Moreover, the SS substrate is exposed to the atmosphere, the driving force being the reduction in surface free energy of the oxidized metal surface by the adsorption of air-born carbonaceous materials [Applied Surface Science 403 (2017) 240-247].
Q3) When photoirradiation was carried out, BP layer may be activated and radicals were formed. However, no reaction might be occurred with clean SS substrate. What kind of binding reaction did occur? From viewpoint of this, Figure 1a could not be acceptable.
Response: Initially, the pristine SS substrate was passivated with piranha solution to oxidize the organic residues. The substrate was then cleaned with ethanol, acetone and DI water. Piranha and protic solvents may induce hydroxyl or unsaturated metal oxides. [ACS Applied Materials and Interfaces 4 (2012) 3254-3260]. The phenyl moieties from BP units could readily react with the hydroxyl groups via condensation mechanism [Colloids and Surfaces B: Biointerfaces 126 (2015) 162.168]. The pristine SS surface had functional groups of O-H, C-O, C-H stretching vibrations, as shown in the FTIR spectra (supplementary information Fig. S1). The functional groups on the SS surface may possibly be responsible for the substrate-binding attribute. The details are included in the 1st paragraph of section 3 (p. 6) and section 3.2 (p. 9).
Q4) As Figure 2 and Figure 3 are not considerable. The reviewer recommended that these figures would be deleted. The thickness of the PMPC layer should be added instead of RMS value. Did authors consider that under dry condition, did the PMPC layer collapse or stand up on the surfaceh?
Response: Thank you for your suggestion. We believe that Figs. 2 and 3 are important to show the surface morphology changes after different treatments. The FESEM images show a resolution of ~10 µm and the macrostructure does not seem to differ between samples. The AFM reveals the µm resolution and the surface morphology is smoother especially on MPC modified sample. This result shows the submicron or nanomicron topological changes upon MPC grafting. The details are included in the 2nd paragraph of section 3.1, p. 8.
We wanted to show the MPC smooth surface. We also kept the RMS values.
The MPC layer was estimated to the tens of nanometers thick and the details are included in the 2nd paragraph of section 3.2, p. 10.
The surface topography was analyzed using dry conditions and the MPC layer was collapsed on the surface. The detailed discussion is included in the 2nd paragraph of section 3.3, p. 10.
Q5) Figure 4 is not clear indication. It should move to the supporting data.
Response: Thank you for the suggestion. Figure 4 is moved to supplementary materials (Fig. S3).
Q6) As biological test, bacteria adhesion and cell adhesion were carried out. However, the effects of the PMPC layer on the substrate did not observed significantly. In generally, many reports have been indicated that the PMPC layer can prevent both bacteria adhesion and cell adhesion. Of course the reviewer understand the mode of binding reaction with substrate was different. However, if the surface was covered with the PMPC layer stably, the resistance performance for bacteria and cell adhesion should be the same with the published data. One possibility is unstable of the PMPC layer during culturing period. That is, the reviewer has already pointed out, the binding between BP layer and SS substrate was too weak to maintain the PMPC layer for long term. The authors must make sufficient explanation about this point.
Response: Thank you for your valuable suggestion. The details are included in the 3rd paragraph of section 3.6, pp. 15.

Reviewer 3 Report
The authors presented the obtaining of biocompatible 2-methacryloyloxyethyl phosphocholine polymer grafted onto 316L stainless steel substrate using the photo-induced radical graft polymerization method via benzophenone (BP) and investigation of both antibacterial and cell survival effects. This manuscript shows new and interesting results, and it is well written. However, some aspects must be improved before its publication in Nanomaterials.
- The names of microorganisms must be written in italic in the whole manuscript.
- The abbreviations must be defined at their first use in the Manuscript (ex. MTT).
- Please provide the name of the software used for the statistical analysis.
- I suggest to include in the Figure 1b – “Antibacterial effect and cell attachment”.
- Why did you select the time interval of 12 h for the incubation with bacteria? Did you try also other time intervals?
- I did not understand the statistical analysis performed for each graph. Could you explain to which group did you compare the other samples in Figures 6, 8 and 9?
- Please explain what represents the control in Figures 8 and 10.
Author Response
Revision 3
English language and style
( ) Extensive editing of English language and style required
( ) Moderate English changes required
(x) English language and style are fine/minor spell check required
( ) I don't feel qualified to judge about the English language and style
Response: The language is checked and corrected.
Comments and Suggestions for Authors
The authors presented the obtaining of biocompatible 2-methacryloyloxyethyl phosphocholine polymer grafted onto 316L stainless steel substrate using the photo-induced radical graft polymerization method via benzophenone (BP) and investigation of both antibacterial and cell survival effects. This manuscript shows new and interesting results, and it is well written. However, some aspects must be improved before its publication in Nanomaterials.
Thank you very much for your positive response to our work. We improved the revised manuscript based on your suggestion.
Q1) The names of microorganisms must be written in italic in the whole manuscript.
Response: Thank you for pointing this out. The microorganism was written in italics in the revised manuscript.
Q2) The abbreviations must be defined at their first use in the Manuscript (ex. MTT).
Response: This is corrected in the entire manuscript.
Q3) Please provide the name of the software used for the statistical analysis.
Response: We used MegaStat software (a MS Excel add on) to calculate the significant difference in multiple comparisons. The details are included in the section 2.8, p.6.
Q4) I suggest to include in the Figure 1b – “Antibacterial effect and cell attachment”.
Response: Thank you for your suggestion. Figure 1(b) is revised.
Q5) Why did you select the time interval of 12 h for the incubation with bacteria? Did you try also other time intervals?
Response: The authors fully agree with the Reviewer’s professional viewpoint about the incubation time effect on antibacterial activity. Actually, other investigators have examined the susceptibility of S. aureus in mid- and late-exponential phase (6 h and 12 h) and the stationary phase (24 h) to the antimicrobial agents found that the bacterial cells are most sensitive to the test substances in the late-exponential phase of culture [International Journal of Food Microbiology 102(2) (2005) 213-220]. As actively growing bacterial cultures become inactive, the cells tend to be more resistant and difficult to respond to antimicrobials, which may be unsuitable for use in the evaluation of antibacterial effects. Therefore, the authors selected the 12 h experimental condition (rather than other time intervals) for the incubation with bacterial cultures to more accurately investigate the antibacterial effects. Following the Reviewer’s valuable advice, the authors have added these descriptions to the text to clarify our antibacterial test. The details are included in the 1st paragraph of the section 2.6, p. 5.
Q6) I did not understand the statistical analysis performed for each graph. Could you explain to which group did you compare the other samples in Figures 6, 8 and 9?
Response: Data with different letters mean they are significantly different on multiple comparison at α=0.05 using Tukey analysis. The figures are revised and the details are included in the figure captions.
Q7) Please explain what represents the control in Figures 8 and 10.
Response: The bacteria and cell control groups are mentioned in the 1st paragraph of section 2.6 and 2.7, p. 5. The details are included in the figure captions.
Round 2
Reviewer 1 Report
The paper was revised according to the comments.